# Three Layered Architecture for Driver Behavior Analysis and Personalized Assistance with Alert Message Dissemination in 5G Envisioned Fog-IoCV

**Mazen Alowish [1,*], Yoshiaki Shiraishi [1], Masami Mohri [2] and Masakatu Morii [1]**

[1] Department of Electrical and Electronic Engineering, Kobe University, Kobe 657-8501, Japan; zenmei@port.kobe-u.ac.jp (Y.S.); mmorii@kobe-u.ac.jp (M.M.)
[2] Department of Electrical, Electronic and Computer Engineering, Gifu University, Gifu 501-1193, Japan; mmohri@gifu-u.ac.jp
[*] Correspondence: 158t802t@stu.kobe-u.ac.jp

**Abstract:** The Internet of connected vehicles (IoCV) has made people more comfortable and safer while driving vehicles. This technology has made it possible to reduce road casualties; however, increased traffic and uncertainties in environments seem to be limitations to improving the safety of environments. In this paper, driver behavior is analyzed to provide personalized assistance and to alert surrounding vehicles in case of emergencies. The processes involved in this research are as follows. (i) Initially, the vehicles in an environment are clustered to reduce the complexity in analyzing a large number of vehicles. Multi-criterion-based hierarchical correlation clustering (MCB-HCC) is performed to dynamically cluster vehicles. Vehicular motion is detected by edge-assisted road side units (E-RSUs) by using an attention-based residual neural network (AttResNet). (ii) Driver behavior is analyzed based on the physiological parameters of drivers, vehicle on-board parameters, and environmental parameters, and driver behavior is classified into different classes by implementing a refined asynchronous advantage actor critic (RA3C) algorithm for assistance generation. (iii) If the driver's current state is found to be an emergency state, an alert message is disseminated to the surrounding vehicles in that area and to the neighboring areas based on traffic flow by using jelly fish search optimization (JSO). If a neighboring area does not have a fog node, a virtual fog node is deployed by executing a constraint-based quantum entropy function to disseminate alert messages at ultra-low latency. (iv) Personalized assistance is provided to the driver based on behavior analysis to assist the driver by using a multi-attribute utility model, thereby preventing road accidents. The proposed driver behavior analysis and personalized assistance model are experimented on with the Network Simulator 3.26 tool, and performance was evaluated in terms of prediction error, number of alerts, number of risk maneuvers, accuracy, latency, energy consumption, false alarm rate, safety score, and alert-message dissemination efficiency.

**Keywords:** clustering; driver behavior analysis; alert message dissemination; personalized assistance; E-RSU; fog computing

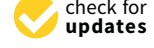



## 1. Introduction

In recent years, the rapid development of intelligent transportation systems (ITS) and the Internet of things (IoT) has contributed a lot to a fast-emerging technology called "Internet of connected vehicles" (IoCV) [1]. Vehicles are connected with each other and with road infrastructure for better transportation and to avoid complications from traffic congestion and road accidents. Car crashes, head-on accidents, fires, and roll-on events can be accurately detected with the assistance of on-board sensors that are deployed in intelligent vehicles [2]. In the broad field of accident prevention and road safety, driver behavior analysis is prominent since a large number of accidents occur due to a lack of attention by drivers [3,4]. In the analysis of driver behavior, it is also necessary to consider

traffic and junction characteristics to increase accuracy [5]. Edge-assisted road side units (E-RSUs) play a vital role in monitoring vehicles [6]. Recently, 5G wireless networks have permitted researchers to cover an enormous number of vehicles to collect real-time data and then analyze the behavior of the drivers [7].

Additionally, current smart systems use alert generation modules in monitoring systems to send immediate alerts to corresponding vehicles [8]. When we look back at the research work on driver behavior, the researchers usually followed the practice of analyzing past events collected from both vehicular and infrastructure communication systems and evaluating the data to compute a safety score for drivers [9]. Moreover, a minimal number of research papers have focused on personalized recommendations for corresponding behaviors of drivers [10]. However, intelligent recommendations are still made on-demand to the driver on the basis of actions the driver makes [11]. To make such recommendations, vehicles, road side infrastructure, and smart wearable devices and sensors worn by drivers must be used to collect information on each individual vehicle's driver [12].

Another main challenge with connected vehicles is the computation of data; vehicles are often not efficient at computing data and thereby offload the task to the cloud [13]. These computations have to be performed at an ultra-low latency. The conventional cloud computing technique cannot be used as it involves transmission delay, and this might cause delayed decisions, which would result in congestion, traffic delays, and accidents [14]. To overcome delays, the computation process has to be distributed to a number of fog nodes in a decentralized manner [15]. These fog nodes are responsible for computing the data for separate regions and sending only the result to the cloud to meet reliability challenges [16]. A fog node has a processing ability higher than that of E-RSUs.

Current advancements in artificial intelligence (AI) such as machine learning, deep learning, and deep-reinforcement learning support massive real-time data processing and decision making [17]. The clustering of vehicles in a network reduces complexity, resulting in reduced time consumption [18]. The assistance provided to the driver should be generated dynamically as traffic increases [19]. The driver should be provided with prior assistance regarding lane changes in order to avoid risks [20,21]. To explore more effective results, a specific choice of algorithm using AI is highly required.

The major aim of this research is to reduce road casualties in an environment by connecting vehicles and by detecting driver behavior. Driver behavior is considered in order to provide intelligent driving assistance to the driver. An alert message is disseminated to the surrounding vehicles that are near a vehicle presenting risks. The behavior of the driver is detected and categorized into several classes so as to assist the driver precisely. The common research issues encountered in this area are:

- Heterogeneous data—To analyze the behavior of drivers, many types of information are needed, including information on drivers' physical states (drowsy, distracted), vehicles' on-board information (accelerating, braking), and environmental information (road conditions, weather conditions). Processing such data dynamically increases complexity.
- Uncertainties—in real-life scenarios, there are many uncertainties in the environment that need to be considered in order to design an efficient assistance module.
- The proposed driver behavior analysis and personalized assistance model are designed in such a way as to overcome these issues. The major objective of this research work is to design a novel system for detecting driver behavior and providing intelligent assistance. The sub-objectives are listed below,
- To minimize the computational overhead during driver behavior analysis.
- To minimize latency in forwarding alert messages.
- To minimize transmission delay in offloading and maximize the resource utilization of fog nod
- To minimize the energy consumption of fog nodes

The main focus of this research work is to provide personalized assistance to drivers and to alert drivers of casualties around them as a step to preventing them from harm. The major contributions of this work are listed below.

- Vehicles in the road environment are clustered by edge-assisted road side units (E-RSUs) using multi-criterion-based hierarchical correlation clustering (MCB-HCC) [22] in order to reduce the complexity in handling a large number of nodes. The information about these vehicles is processed in an attention-based residual neural network (AttResNet) [23] to classify the vehicles on the basis of mobility.
- The driver behavior of the above classified vehicles is analyzed for several drivers, vehicles, and environmental conditions by implementing a refined asynchronous advantage actor critic (A3C) Algorithm. In this process, the driver's current state is analyzed and classified into several classes such as drowsy, distracted, in an emergency, speeding, bad pedaling, and bad steering.
- In case of emergency situations, alert messages are disseminated to surrounding vehicles of the vehicle presenting a risk, which is performed via fog nodes using a jellyfish search optimization (JSO) algorithm [24]. An alert message is also disseminated to nearby optimal fog nodes, and in the case that there are no fog nodes, a virtual fog node is created dynamically by using a constraint-based quantum entropy function.
- Personalized assistance is provided to the driver on the basis of the analyzed behavior of the driver. Assistance is provided one or more times by utilizing a multi-attribute utility model. This assistance facilitates in preventing road casualties. All instances of assistance are stored in a cloud server for long-term assessment of driver behavior.

The rest of the sections in the manuscript are organized as follows. Section 2 shows the literature survey of the previous works which are relatable to the proposed work. Section 3 explains the problem statement which is identified in the previous works. Section 4 represents the proposed work methodology with corresponding mathematical representations, pseudocode and procedure, which is followed by simulation setup, experimental results and comparative analysis between the proposed and existing works. Finally, Section 6 describes the conclusion and future work of the proposed research.

## 2. Related Work

An integrated method for overcoming the problem of road accidents occurring during the transportation of dangerous goods was proposed in [25]. Driver behavior is analyzed by analyzing eight indicators on the basis of the huge amount of data provided by the operating vehicle networked control system. From that, three important indicators, (1) acceleration and deceleration behavior, (2) over speed behavior, and (3) operation stability, are introduced. The genetic algorithm-fuzzy-C mean clustering method (GA-FCM) is used to classify 40 drivers on the basis of behavior parameters. In accordance with the results of driving behavior, the drivers are analyzed on the basis of the indicators. Hence, the proposed method combines statistical analysis and GA-FCM to analyze the behavior of drivers in the transportation of dangerous goods. This will help the respective department train high-risk drivers and thereby control accidents. However, only three indicators are focused on briefly, which is not enough to classify the behavior of drivers, and the number of drivers used for the real-time experiment of this method was not sufficient enough to analyze driver behavior. In [26], the authors proposed a virtual soft sensor based on neuro-fuzzy systems and principal component analysis. The authors proposed this sensor to eliminate the need to install additional hardware in a vehicle, thereby reducing the cost of the vehicle. On the basis of parameters such as speed, acceleration, and inertial measurements, driver behavior is classified into three classes: drowsy, normal, and aggressive. The proposed virtual sensor was validated by carrying out a test with five different drivers and vehicles on two different types of road, that is highway and secondary roads, and the result showed that the virtual sensor has better performance in classifying driver behavior. However, there are only three classes to classify the behavior of drivers, which is not sufficient for

differentiating driver behavior. The parameters used to classify driver behavior did not include any health parameters, which play an important role in analyzing driver behavior.

The authors in [27] proposed a system for driver behavior analysis using a deep convolutional neural network (DCNN) based on biological signals for real-time detection. There are three modules used in the method. The first module is used to detect the behavior of the driver, which is carried out by a Raspberry Pi 3 detached from the vehicle. The second module is used to train the DCNN, and the third module is used to monitor and profile driver behavior for supervision. Thus, the proposed system performs better than other machine learning algorithms by analyzing driver behavior. However, the computations are performed in the cloud, which increases latency and transmission delay. The feedback given by this model is very limited, and no recommendations are given to the driver to change behavior. In [28], the authors proposed an adaptive computation offloading method (ACOM) as a framework for reducing the transmission delay and increasing the resource utilization involved in offloading. To achieve this, a multi-objective evolutionary Algorithm using a decomposition method is used. The algorithm provides D number of optimal strategies to be chosen, in which the best strategy is obtained by normalizing the strategies based on TOPSIS and MCDM. The performance of the obtained strategy was evaluated for different parameters, and the result proved the efficiency of the proposed framework. Nevertheless, the optimal strategy provided by the proposed method is not sufficient in real-world scenarios because once an edge gets overloaded, it is of no use.

A mobile-based application for the analysis of driver behavior is provided in [29], and it is used by getting inputs from a mobile camera, sensors deployed on the vehicle, and audio input from the vehicle. The raw data of the inputs are tracked to find eye openness, mouth openness, head yaw angle, head pitch angle, the vehicle's position and to recognize speech. Through this tracked information, the driver's states of danger can be classified into drowsiness, distraction, high pulse rate, drunk driving, aggressive driving, and stress. Once the driver's behavior is found to be in a dangerous state, recommendations are generated and given to the driver. The proposed method was tested in real time, and it enhanced driver safety and reduced the probability of road accidents. The limitations faced in this work are that the proposed model will not be suitable in remote areas where mobile connectivity can be disturbed. In addition, every time a new driver drives a car, he or she has to install the application on their phone. In [30], the authors proposed an approach to effectively increasing the spreading of information by deploying a latent edge with high potential influence. The strategy is called the "latent-edge-influence strategy" (LEI). Through this strategy, the highest potential influence at which an edge should be introduced is calculated so that it will provide effective performance in spreading information. The proposed method was compared with the degree-product strategy (DP) and eigenvector centrality product (ECP). A monitoring and coaching strategy for reducing road accidents caused by vehicles transporting heavy goods was proposed in [31]. Using this strategy, the authors compared the vehicles of two companies, A and B, to calculate the increase in performance by implementing the strategy. The time period for comparing the vehicles was one year, in which the first eight months were considered as the baseline period in which parameters such as harsh braking, harsh cornering, and over speeding were noted. The next time period of four months was termed the intervention period. During this period, Company A deployed both monitoring and coaching, whereas Company B deployed only monitoring. By calculating the above-mentioned parameters during the intervention period for both companies, it was proved that the proposed strategy performs better in reducing road accidents and improving driver safety. However, the strategy uses only cameras to detect when the driver is in a state of emergency, which will not be accurate. The parameters used in this strategy will not be sufficient enough to differentiate the causes of accidents. In addition, no real-time recommendations are given to the driver, which would be more effective than coaching. In [32], the authors proposed a fuzzy cluster analysis based on risk sensitivity and a judgment threshold for two different age groups of drivers. The risk level was evaluated by integrating both data obtained for expert drivers and the drivers

of the different age groups. Fisher discriminant analysis was used to classify drivers into discriminant models. The analysis showed that the risk sensitivity and judgment threshold are better for drivers of the younger age group, whereas the drivers of the elderly age group ignore risks due to inefficient judgment of the threshold level. This showed that the younger drivers adapted slowly to the driving simulation system. However, the analysis was carried out with a low number of drivers; if it were performed under a real-world scenario, the accuracy would be decreased, and the proposed model would not detect the behavior of drivers, which would further affect the performance.

A hybrid convolutional neural network framework was proposed in [33] by combining residual neural network 50, Inception version 3, and Xception. Transfer learning is used to extract the features of drivers. These features are concatenated to obtain useful information. The proposed framework is trained to ignore the non-distractive behavior of drivers. Ten classes of typical driver behavior were tested, and the features were highlighted by using class activation mapping. The result showed that the proposed framework was able to detect distractions in driver behavior up to 97%, thereby reducing road accidents. The proposed framework will detect only behavior indicating that drivers are distracted, but apart from distractions, many factors cause road accidents. The proposed framework also does not provide any recommendations to the driver. In [34], the authors proposed a novel framework called "driver model adaptation" (DMA). A transfer learning-based approach was developed to find the data of various drivers and to adapt the obtained data to the dataset of a target driver. The name of this approach is dynamic time warping with local procures analysis (DTW-LPA). Once the adaptation is completed, Gaussian mixture regression is used to train the target driver's model. The proposed model was validated under various experiments with the data collected. The result showed that the proposed framework performs well with more accuracy. It does not detect the behavior of each driver; it uses only the data of other drivers to train the target driver model. It does not provide any real-time recommendations to the drivers based on their behavior.

An assistant system was proposed in [35] that uses a bidirectional long short term memory (Bi-LSTM) network to detect the turning behavior of a vehicle with derived parameters such as lateral velocity, lateral acceleration, and heading angle and predicted parameters such as lateral position, longitudinal position, speed, and acceleration of the vehicle, which are predicted by using an online auto regressive integrated moving average (ARIMA) algorithm. The next generation simulation (NGSIM) was used to validate the proposed system, and the result showed that turning behavior was predicted at an accuracy of 94% before initializing turning. However, the assistant system predicts only the turning behavior of a driver, which will not reduce road accidents caused by other situations. In [36], the authors proposed a driver behavior model based on the whale optimization algorithm—restricted Boltzmann machine (WOA-RBM) method. A restricted Boltzmann machine used in deep learning is used to mimic the behavior of a driver. The optimal parameters for the RBM are determined by the whale optimization algorithm. Finally, the output of the proposed method is used to control a vehicle by applying acceleration and deceleration. The proposed driving model was tested and validated in MATLAB, and the results showed that it performed better in prediction and had an accuracy of 90%. Nevertheless, it assists only in acceleration and deceleration and does not provide any assistance for improper handling of a vehicle. It also does not give any recommendations to the driver based on their behavior.

A driving safety field (DSF) model was proposed in [37] to predict pedestrians and to avoid the collision of vehicles and pedestrians. This model consisted of five blocks. The first block is an input block that gathers information on pedestrians, vehicles, and the road environment. The second block is for pedestrian trajectory prediction, in which a dynamic Bayesian network (DBN) is used to predict the intention of pedestrians. The third block is for vehicle trajectory prediction, in which the vehicle is assumed to travel at a constant velocity and invariant yaw angle. The fourth block is used to assess risk, and the fifth block provides a pedestrian risk value. A Monte-Carlo experiment was used to

simulate the vehicle driving process, and the result showed that the proposed model could be implemented in autonomous vehicles to make decisions for safe driving. However, it does not predict the risk of collision caused by adjacent vehicles, which will also lead to accidents, and it does not provide any recommendations to the driver.

The authors in [38] proposed an adaptive cruise control method for analyzing the behavior of adjacent vehicles while lane changing to avoid collisions. First, parameters such as the acceleration and velocity of an adjacent vehicle are estimated. After this, the trajectory of the adjacent vehicle is predicted. Risk assessment and risk minimization are carried out by the proposed method. The method was evaluated in a real-time situation, and the performance of each stage of the method was analyzed. The result showed that the method reduces collisions, thereby improving driver safety. However, it does not give any recommendations to the driver based on speed, braking, and control of steering. In [39], the authors proposed a risk prediction method used during lane changes. The physiological information of the driver is obtained from sensors fixed on the driver. The hidden Markov model (HMM) is used to reduce the driving risk with the information obtained from the sensors. Physiological information such as ECG factors and eye movement factors are evaluated before predicting the driver's risk. The dynamic factors of vehicles such as average speed and acceleration are also evaluated to reduce false input data. The proposed method was tested in MATLAB, and the result showed that this method predicted the driver's risk more accurately. The Table 1 above represented the research gaps in the state-of-the-art works in which parameters, Algorithm, and limitations of the existing works are given.

**Table 1.** Research Gaps in Previous Works.

| References | Objective | Input Parameters | | | Algorithm/Setup Used | Features Extracted/ Working | Limitations |
|---|---|---|---|---|---|---|---|
| | | Physiological | Vehicular | Environment | | | |
| [25] | Behavior analysis | × | ✓ | × | Genetic algorithm-fuzzy-C mean clustering (GA-FCM) | Acceleration and deceleration Over speed Operational stability | Lack of consideration of adequate information results in inefficient behavior identification |
| [26] | Behavior analysis | × | ✓ | × | Neuro-fuzzy system with principal component analysis (FIS-PCA) | Speed Acceleration Inertial measurements | Driver is classified as drowsy without considering physical information, which increases false alarm rate. |
| [27] | Behavior analysis | ✓ | × | ✓ | Deep convolutional neural network (DCNN) | Heart rate Blood pressure Driver action | Feedback provided by this model is very limited, and latency is high as processing is performed in cloud. |
| [28] | Edge offloading | - | - | - | Multi-objective evolutionary algorithm based decomposition (MOEA/D) | Generates optimal solutions to increase resource utilization and reduce delay | Not suitable for large-scale, real-time environment in which increased computation results in overloading. |
| [29] | Behavior analysis | ✓ | ✓ | × | Smartphone mobile application | Forehead color Head movements Vehicle position Speed Speech input | This assistance gets disturbed due to connectivity problems in remote areas, and installation of application is required for new drivers. |

**Table 1.** *Cont.*

| References | Objective | Input Parameters | | | Algorithm/Setup Used | Features Extracted/ Working | Limitations |
|---|---|---|---|---|---|---|---|
| | | Physiological | Vehicular | Environment | | | |
| [30] | Edge offloading | - | - | - | Latent-edge-influence strategy (LEI) | Effective performance in spreading information based on SIR information | Potential influence calculated for deployment of latent edge is not sufficient as it does not consider communication delay. |
| [31] | Driver monitoring | × | ✓ | ✓ | Analysis of variance (ANOVA) | Harsh breaking Harsh cornering Over speeding | Harsh conditions cannot be identified accurately as this approach uses only camera sensor. |
| [32] | Risk evaluation | × | ✓ | × | Fuzzy cluster analysis (FCA) | Velocity Brake pedal Wheel steering Longitudinal acceleration | Risk assessment is based on age group only; lack of consideration of driver's physical information provides inaccurate results. |
| [33] | Behavior analysis | × | ✓ | × | Hybrid convolutional neural network framework combining residual neural network 50, Inception version 3, and Xception | Face expressions Eye movement Hand gestures Speech features | Distracted behavior of driver is detected using only image and audio inputs, but there are other physical factors that contribute to distraction. |
| [34] | Driver model analysis | × | ✓ | × | Dynamic time warping with local procures analysis (DTW-LPA) | Steering angle Front wheel angle | Driver model adaptation technique does not consider driver behavior during uncertain situations |
| [35] | Behavior analysis | × | ✓ | × | Auto regressive integrated moving average (ARIMA) | Lateral velocity Lateral acceleration Heading angle | This system predicts only turning intention, which will not fully reduce road casualties. |
| [36] | Behavior analysis | × | ✓ | ✓ | Whale optimization algorithm-restricted Boltzmann machine (WOA-RBM) | Road geometry Weather condition Speed Energy | Model is based on acceleration and deceleration of vehicle but does not consider steering properties. |
| [37] | Collision avoidance and risk assessment | × | ✓ | × | Advanced adaptive cruise control system | Position Speed | Provides collision control and risk assessment only in cut-in situations. |
| [38] | Behavior analysis and risk prediction | ✓ | ✓ | × | Hidden Markov model (HMM) | ECG data Eye movement data Vehicle speed and dynamics | Risk prediction accuracy does not consider traffic conditions, which reduces accuracy. |

## 3. Problem Statement

Driver behavior analysis and vehicle motion prediction are the major aspects of road safety provisioning. Many research works have been focused on road safety and accident prevention for Internet of connected vehicle (IoCV) environments. However, the existing work focus on either behavior analysis or motion prediction for accident detection. In particular, driver behavior is analyzed only by detecting vehicle parameters and road

information. Recommendations to the driver based on their behavior are not provided accurately, and there are more delays. Additionally, the following research problems in driver behavior analysis are encountered.

- High false-alarm rate: Due to limited consideration of parameters from the driver and surroundings, the prediction of accidents is not accurate and leads to high traffic congestion in both urban and rural areas and re-routing of nearby vehicles. For accurate driver-behavior detection, driver health parameters (pulse rate, EEG rate, respiration rate, and heart rate), vehicle-motion-detection parameters (speed, longitudinal acceleration, yaw angle rate, gyroscope, and magnetometer), and surrounding factors (traffic density, accidents, weather conditions, and pedestrians) are required. Lack of these important factors affects accuracy.
- Lack of optimal assistance for drivers: Recommendations are provided to the driver only on the basis of vehicle motion detection, which is not sufficient enough to assist drivers. The state of driving is not analyzed to give recommendations to drivers. To overcome this, driver states such as drowsiness, distraction, and emergency should be detected.
- High latency: In the existing works, latency is observed in two fields. Computing tasks in a cloud environment increases time consumption and thereby increases the probability of accidents occurring. Even when computations are carried out in an edge server, the higher number of vehicles in a real-time scenario makes it time consuming.

A clustering method for grouping vehicles on the basis of behavior was proposed in [40]. Integrating a deep learning algorithm with clusters, the authors proposed a location prediction algorithm to localize vehicles in the future. Roads are considered as small portions to easily locate vehicles; hence, long-term predictability is possible. Stacked auto-encoding is used to predict the location of vehicles, which are mapped to the road. The result of a simulation showed that the proposed method is accurate in prediction. It can help vehicles select better routes and reduce traffic congestion. The major problems faced in this research work were:

- Clustering can be performed effectively, however, if the number of samples increases, scaling issues will occur.

The stacked auto-encoder may work perfectly with a training set but fail miserably when samples run out. It is more sensitive to input errors different from those in the training set, which could cause massive errors in an auto encoder, and it might have to be retrained from scratch. It focuses only on the quantity of information rather than the quality.

Research solutions: The efficiency of the clustering can be improved by using multi-criterion-based hierarchical correlation clustering (MCB-HCC). AttResNet is used to focus on the quality of the information and to avoid retraining the network.

The authors in [41] proposed an offloading method for carrying out the computations of connected vehicles at an ultra-low latency by deploying multi-access edge computing servers along with road side units (RSU's). The selection of MEC servers by fast moving vehicles is carried out by a deep Q learning algorithm, which will solve the frequent handover problems. The centralized management of the vehicular network is taken care of by a software-defined network.

The result of a simulation showed that the proposed method has better performance and adaptation than the older offloading methods. The issues faced in this paper were as follows.

- When higher requirements are put forward for timely task processing, which involves a large number of computations, task offloading faces obstacles.
- Even though the offloading is performed with ultra-low latency, the offloading quality (i.e., storage, deadline delay constraints, and energy efficiency) is not good and should be improved.
- Lack of environmental factors leads to mobile edge server failure, so offloading accuracy is decreased.

Research solutions: Physical and virtual fog deployment in IoCV environments can reduce response time and latency. The deployment of servers for the highest potential influence will increase the quality of offloading.

A driver behavior model that analyzes the behavior of drivers was proposed in [42]. It measures information such as vehicle data, environmental data, and socio-demographic data. A Gaussian mixture model-universal background model (GMM-UBM) is used to construct an individual model for each drive to store individual behavior profiles. The drivers are given scores based on an analysis performed by the model. The scores are given with respect to the score of the safest driver, which is considered from the historical records of the drivers or by calculating the average normal distribution of all the drivers. Based on the scores, the drivers will get recommendations to mimic the behavior of the safest driver. Through this method, risky drivers can adopt better driving behavior, thereby reducing the risk of road accidents. However, this method has several problems, which are mentioned below.

- A different level of connectivity for longer tests with more vehicles within other networks cannot be calculated.
- The data library is not sufficient because it does not have more trajectory data in addition to other types of data describing driver situations, particularly weather conditions and driving behavior, and real-time updating of the data library should be implemented.
- The health parameters of drivers are not considered, and they are important in emergencies.

Research solutions: Driver behavior can be analyzed by considering driver health parameters, vehicle motion detection parameters, and surrounding factors. The driver analysis is performed by a refined asynchronous advantage actor critic (A3C) algorithm. An intelligent driver assistant was proposed that accompanies the driver by providing real-time audio-visual alerts in [43]. The proposed system uses both road accident risk map analysis and on-board telemetry, which are integrated based on fuzzy logic. The on-board telemetry system monitors the driver by collecting data such as speed, bad pedaling, and bad steering. The proposed system analyzes a certain amount of accident data through various parameters and estimates the road risk level, thereby identifying which road is riskier than the other. Through this proposed method, the efficiency of the driver gets improved. The drawbacks of this approach are below.

The assistance given by the car is very limited as it does not give any real-time warnings.

- Regarding drivers not only having the risk of an accident for a particular road section but also sub-section or intersection, such details are not given.
- Environment characteristics such as weather conditions are not observed in the road accident risk map analysis but do have an impact on road accidents

Research solutions: Personalized assistance can be provided in real-time on the basis of the current state of the driver. Assistance is provided one or more times through a multi-attribute utility model. This model is accurate and reduces the latency in giving assistance.

A beaconless approach for data communication between two vehicles was proposed in [44] to overcome the wastage of bandwidth caused by beacon packets. The packets to be transmitted are rescheduled on the basis of the forwarding probability, which is decided by factors such as distance, angular orientation, moving direction, and buffer delay. Among the rescheduled packets, the highest priority packet is transmitted first. Hence, the rebroadcasting of packets is saved. The results of a simulation showed that the proposed system is better in performance metrics such as reachability and average delay. This research work has disadvantages as follows.

- Fuzzy logic is not always accurate; hence, the results are based on assumption. This increases the number of rules for each vehicle to check to make decisions, which is time consuming for the vehicle.
- If there are no neighboring vehicles, the data packets are sent to road side units, and selfish nodes should be identified to eliminate unnecessary packet drops.

- Research solutions: The jellyfish search optimizer can be used to select risky vehicles, which will increase the accuracy. All alert messages are stored in the cloud for further assistance

## 4. Proposed Work

In this research work, we concentrate on driver behavior analysis, generating alert messages to the surrounding vehicles, and also assisting the driver. For that, a novel three layered architecture for driver behavior analysis and a personalized assistance system are designed as shown in Figure 1. The proposed work is organized in three layers as follows.

- Layer 1 (connected vehicles): This layer consists of intelligent connected vehicles that have onboard sensors to collect information such as speed, longitudinal acceleration, yaw angle rate, gyroscope information, and magnetometer information. The drivers inside the vehicles are deployed with smart wearable sensors to gather information such as pulse rate, EEG rate, respiration rate, and heart rate. The collected data are sent to the upper layers through communication technologies. Layer 1 also includes edge-assisted road side units (E-RSUs) to speed up the process.
- Layer 2 (fog computing): This layer consists of distributed fog nodes, which are responsible for monitoring and handling separate regions in Layer 1. The nodes are connected to an E-RSU and have a computation ability greater than it.
- Layer 3 (cloud server): This is the uppermost layer in the architecture and consists of a cloud server that continuously collects and maintains the information provided by Layer-1 and Layer-2 devices.

All of these layers are combined and work to analyze driver behavior and provide personalized assistance to the driver. For that, this work presents multiple contributions that are explained as follows.

### 4.1. Cluster-Based Vehicle Motion Detection

In IoCV, vehicles are connected to each other through V2V communication. To differentiate each vehicle, each has a unique vehicle ID. Similarly, the E-RSU for each road segment have their own unique ID. Roads are considered as road segments to extract the geographical information accurately. The vehicles in an environment are represented as $V = \{V1, V2, V3 \dots Vn\}$. Vehicles travel on the road for a particular time period, and E-RSUs that the vehicle passes are R= {R1, R2, R3 ... Rn}. This information, along with factors such as velocity, speed, latitude and longitude, and the moving direction of the vehicle, are used to form non-overlapping clusters of vehicles from which vehicles will be classified on the basis of mobility. The clustering is performed by E-RSU using MCB-HCC in which the correlation between two vehicles is determined by the distance of correlation. This distance is computed on the basis of the dimensionality of correlation of each vehicle. The covariance matrix $H_V$ of vehicle $V1$ can be formulated as follows.

$$H_{V1} = \frac{1}{|NN_j(V1)|} \cdot \sum_{Q \in NN_j(V1)} (Q - \overline{Q}) \cdot (Q - \overline{Q})^T \tag{1}$$

Here, *j* belongs to a set of vehicles, and $H_{V1}$ of vehicle $V1$ with respect to *j* is created by the *j* nearest neighbor of $V1$ $(NN_j(V1))$. $\overline{Q}$ Is the centroid of $NN_j(V1)$. The dimensionality of correlation $\mu_{V1}$ of vehicle $V1$ can be computed as follows.

$$\frac{\sum_{i=1}^{u} eg_i}{\sum_{i=1}^{d} eg_i} \geq \gamma \tag{2}$$

Here, *u* is the lowest number of eigenvalues $eg_i$, and $\gamma$ refers to the percentage of the total variance. The matrix of correlation similarity is represented as follows.

$$\hat{S}_{V1} = E_{V1} \hat{B}_{V1} E_{V1}^T \tag{3}$$

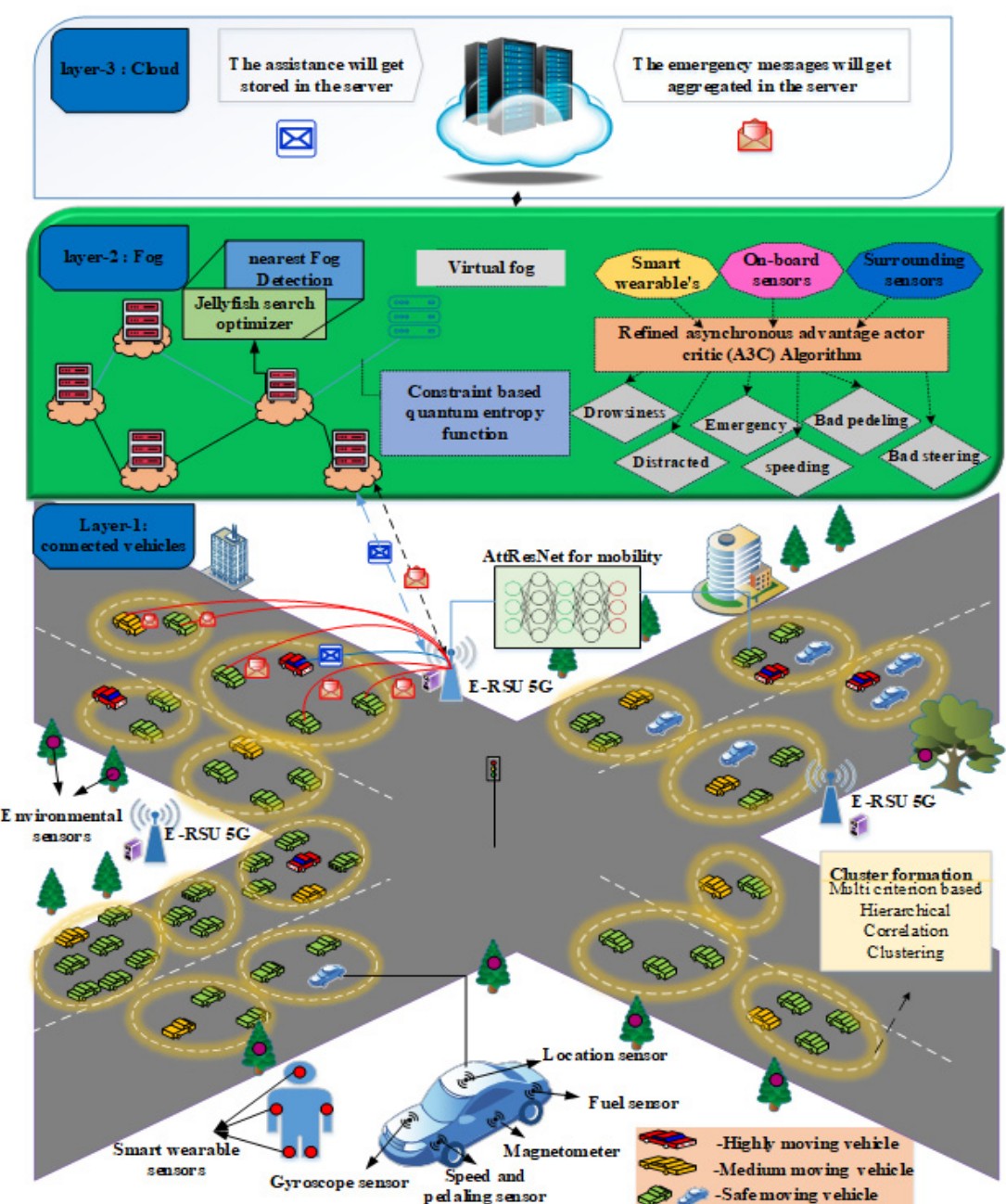

**Figure 1.** Three Tier Architecture.

In Equation (3), $E_{V1}$ denotes the eigenvector matrix of vehicle $V1$ and $\hat{B}_{V1}$ is the eigenvalue matrix formed by the following

$$\hat{eg}_i = \begin{cases} 0, & if\ i \leq \mu_{V1} \\ 1, & otherwise \end{cases} \tag{4}$$

The local distance of correlation of vehicle $V1$ to vehicle $V2$ can be formulated as follows.

$$locD_{V1}(V1,V2) = \sqrt{(V1-V2)^T \cdot \hat{S}_{V1} \cdot (V1-V2)} \tag{5}$$

Here, $locD_{V1}(V1,V2)$ is the Euclidean distance between $V1$ and $V2$, in which $\hat{S}_{V1}$ is taken as weight. However, $locD_{V1}(V1,V2)$ is equal to the distance between vehicle $V2$ and

the correlation hyperplane of the neighbors of $V1$. If $V2$ is present in the hyperplane of $V1$, then we have the following.

$$locD_{V1}(V1, V2) = 0 \tag{6}$$

The dimensionality of correlation between two vehicles $V1$ and $V2$ is indicated as $\mu(V1, V2)$, and the distance of correlation between two points, $V1$ and $V2$, is denoted as follows.

$$CD(V1, V2) = (\mu(V1, V2),\ D(V1, V2)) \tag{7}$$

---

**Algorithm 1:** Distance of correlation

Initialize Vehicles $(V)$
Determine Covariance matrix $H_V$
Compute dimensionality correlation of $V1$ *and* $V2$
Calculate $\breve{B}_{V1}$ from $e\breve{g}_i$**;**
Eigenvector matrix of $V1$ be $E_{V1}$**;**
The dimensionality of correlation of $V1$ is $\mu_{V1}$**;**
Calculate $\breve{B}_{V2}$ from $e\breve{g}_i$**;**
Eigenvector matrix of $V2$ be $E_{V2}$**;**
The dimensionality of correlation of $V2$ be $\mu_{V2}$**;**
Compute $locD_{V1}(V1, V2)$**// Euclidean distance**
For each strong eigenvector $V2_i \in E_{V2}$**do**
    If $V2_i{}^T V2_i - V2_i{}^T E_{V1} \breve{B}_{V1} E_{V1}^T V2_i > \Delta^2$ ***then***
        Adjust $(E_{V1},\ \breve{B}_{V1},\ V2_i,\ \mu_{V1})$;
        $\mu_{V1} = \mu_{V1} + 1$;
For each strong eigenvector $V1_i \in E_{V1}$ do
    If $V1_i{}^T V1_i - V1_i{}^T E_{V2} \breve{B}_{V2} E_{V2}^T V1_i > \Delta^2$ ***then***
        Adjust $(E_{V2},\ \breve{B}_{V2},\ V1_i,\ \mu_{V2})$;
        $\mu_{V2} = \mu_{V2} + 1$;
$CD(V1, V2) = (\mu(V1, V2),\ D(V1, V2))$
$CD = \mathbf{max}(\mu_{V1},\ \mu_{V2})$;
Return $(CD, \mathbf{D_{eucid}}(V1,\ V2))$;

---

The Algorithm 1, is for computing the correlation distance between two vehicles $V1$ and $V2$ is described in, in which the eigenvalue matrix, dimensionality of correlation, and eigenvector matrix of both vehicles are computed to derive the distance of correlation $CD$ as output. Computation of correlation distance to articulate the relationship between the behaviors of the two vehicles to form clusters. The higher the value of $CD$, the more the vehicles are related, which means that $V1$ and $V2$ can form a cluster. $CD(V1, V2) \leq CD(Va, Vb)$ is possible when the following condition is satisfied.

$$\mu(V1, V2) < \mu(Va, Vb). \tag{8}$$

This means that the dimensionality of correlation of $V1$ and $V2$ should be lower than the dimensionality of correlation of $Va$ and $Vb$.

$$\mu(V1, V2) = \mu(Va, Vb),\ D(V1, V2) \leq D(Va, Vb) \tag{9}$$

Here, $D(V1, V2)$ and $D(Va, Vb)$ represents the distance between respective vehicles. The reachability of correlation can be expressed as follows.

$$CR_\lambda(Vn, V1) = \max(CD(Vn, Vm),\ CD(Vn, V1)) \tag{10}$$

Here, $CR_\lambda(Vn, V1)$ denotes the reachability of vehicle $V1$ from $Vn$. $Vm$ is the $\lambda$ nearest neighbor of $Vn$. The pseudo code for MCB-HCC is provided above from which the vehicles are formed as a cluster.

**Pseudo code:** MCB-HCC ($\Im$, $j$, $\lambda$, $\gamma$, $\Delta$)

For each $V1 \in \Im$ do
    Compute $\breve{B}_{V1}$, $E_{V1}$;
    // queue priority **qp** is arranged to $CR_\lambda$
For each $V1 \in \Im$ do
    $V1.CR = \infty$;
    Insert $V1$ into **qp**;
While $qp \neq \phi$ do
    $Vn qp.\textbf{next}()$;
    $Vm\ \lambda$ ***nearest neighbor of*** $Vn$;
    For each $V1 \in qp$ do
**new cr** $=$ **max**$(CD(Vn, Vm), CD(Vn, V1))$;
$qp.\textit{update}(V1, new\_cr )$;

The information obtained from the clusters of vehicles is used to train AttResNet. These features are represented as $(f_1; \ldots; f_z)$ which can be formulated as follows.

$$f_{whole} = [f_1; \ldots; f_z] \tag{11}$$

The equations can be computed as follows.

$$\varepsilon = \rho(B f_{whole} + w) \tag{12}$$

$$\begin{aligned} \vartheta &= \varepsilon \otimes f_{whole} \\ &= [at_1 * f_1; \ldots; at_z * f_z] \end{aligned} \tag{13}$$

Here, $B$ and $w$ are weights, $\rho$ indicates a sigmoid function, and $\otimes$ denotes feature-wise multiplication. However, $\vartheta$ indicates a weighted vector that is let into the regression layer for the purpose of mobility prediction. Here, the sigmoid function is utilized to achieve higher performance. Figure 2 depicts the structure of AttResNet, which consists of an attention layer, noise layer, dense layer, residual layer, and drop-out. In the attention module, the attention value ($at_v$) depicts the importance of a feature ($f_v$) from the available features ($f_{whole}$). The normal distribution of noise in the noise layer is $\mathbb{N}(0, 0.04)$. Overfitting is eliminated by utilizing the drop-out layer. The dense layer uses SELU activation, which can be expressed as follows.

$$SELU(p) = \delta \begin{cases} p, & if\ p > 0 \\ \varepsilon e^p - \varepsilon\ if\ p \leq 0 \end{cases} \tag{14}$$

The loss function for the purpose of training the model can be formulated as follows:

$$L = \frac{1}{N} \sum_{v=1}^{N} (q_v - \hat{q}_i)^2 + \gamma \sum_{u=1}^{M} (\theta_u)^2 \tag{15}$$

Here, the total number of samples is N. The true classification value is denoted as q, and the predicted value is denoted as $\hat{q}$.

The AttResNet model predicts the motion of a vehicle not only by simply classifying vehicles but by interacting with the features by inferring the complexity. From a deep analysis, E-RSU classifies vehicles on the basis of mobility into three classes: high moving, medium moving, and safe moving. Then, this data are set to Layer 2 for further analysis.

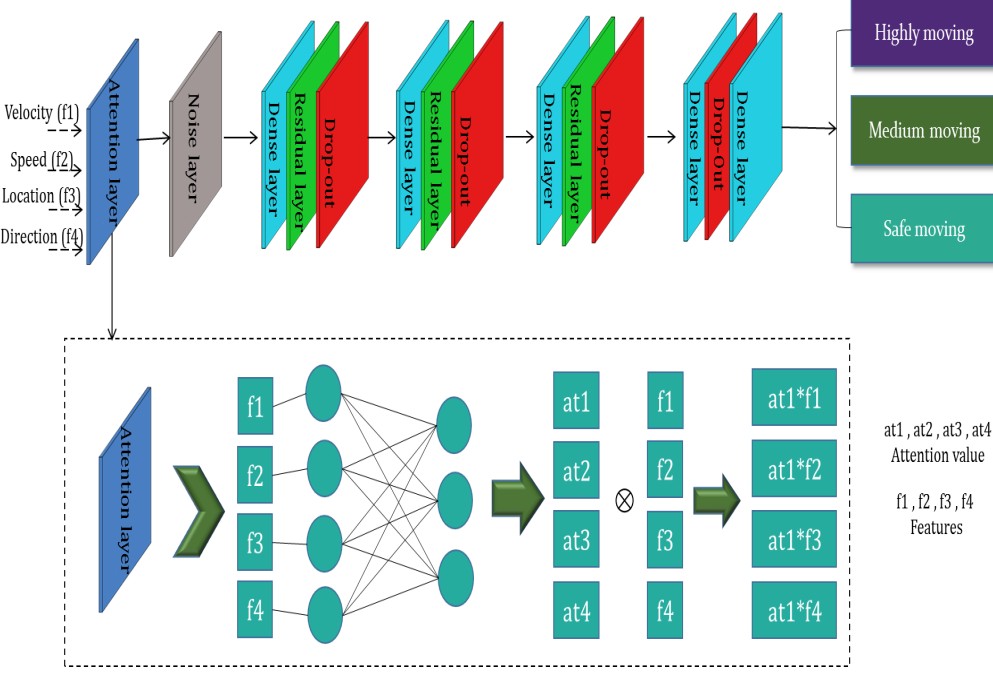

**Figure 2.** AttresNet Model.

### 4.2. Driver Behavior Analysis and Prediction

Make a decision regarding alert generation and personalized assistance. The A3C Algorithm takes heterogeneous data of input such as pulse rate, EEG rate, respiratory rate, and heart rate, which are collected from smart wearable sensors. These inputs are normalized to reduce the redundant data and the normalized data are considered for analysis of driver behavior. In addition, considering vehicle information such as speed, longitudinal acceleration, yaw angle rate, position, and moving direction from on-board sensors and surrounding factors such as traffic density, accidents, and weather condition. These factors are taken as state $(\xi)$ attributes for which the driver's current state, i.e., drowsy, distracted, in an emergency, speeding, bad pedaling, and bad steering, is detected as an action $(\forall)$. The current state is accurately detected by achieving the optimal policy. The proposed Algorithm consists of two entities, an actor network and critic network. The actor network focuses on achieving the optimal policy, whereas the critic network criticizes the performance of the actor network to increase its efficiency. The actor network is represented as $\pi_\theta(\forall|\xi)$, and the critic network is expressed as $C_\phi^\pi(\xi)$, in which $\phi$ is a parameter of the critic network. The approximated $Q$ function can be formulated as follows:

$$Q^\pi(\xi, \forall) \approx \varrho' + \zeta C^\pi(\xi') \approx \varrho' + \zeta C_\phi^\pi(\xi') \tag{16}$$

Here, $\varrho$ is the reward function for each action taken by the actor network, $\xi'$ indicates the next state, and $\zeta$ is the discount factor. The advantage function can be formulated as follows.

$$AD^\pi(\xi, \forall) = Q^\pi(\xi, \forall) - C^\pi(\xi) \approx \varrho' + \zeta C_\phi^\pi(\xi') - C_\phi^\pi(\xi) \tag{17}$$

Though A3C is a policy gradient approach, the critic network is trained on the basis of the value-based approach, which can be represented as follows.

Once the E-RSU classifies vehicle into high moving and normal moving, the fog node triggers the A3C Algorithm to determine the current state of the driver in order to:

$$C^\pi(\xi) = E_{\forall \sim \pi(\forall|\xi)} E_{\xi' \sim p(\xi'|\xi, \forall)} \left[ \varrho' + \zeta C^\pi(\xi') \right] \tag{18}$$

The target value is computed for each update by utilizing present approximation, which can be expressed as follows.

$$x = \varrho' + \zeta C_\phi^\pi(\xi') \tag{19}$$

The pseudo code for the proposed A3C Algorithm is presented below, in which the working process for a single step is explained in a brief manner.

---

**Algorithm A3C:** Pseudo code

---

$C_\phi^* -$ ***critic network***, $\pi\theta -$ ***actor network***, $\eta -$ ***critic scaling loss***, $\beta -$ ***batch size***, SGD optimizer

Randomly initialize $\phi, \theta$**;**

Utilize $\pi(\theta)$ to obtain roll-out size of $\beta$**;**

Advantage estimation is computed for each transition (***tr***) using Equation (17)

Target formulation using Equation (19)

Critic loss calculation:

$$L = \tfrac{1}{\beta} \sum_{tr} \left( x(\boldsymbol{tr}) - C_\phi^\pi \right)^2$$

Critic gradient computation:

$$\nabla^{critic} = \tfrac{\partial L}{\partial \phi}$$

Actor gradient computation:

$$\nabla^{actor} = \tfrac{1}{\beta} \sum_{tr} \nabla_\theta log \pi_\theta(\forall|\xi) AD^\pi(\boldsymbol{tr})$$

Calculate gradient descent using $\nabla^{actor} + \eta \nabla^{critic}$

---

Table 2 represents the normal range of input parameters for the purpose of driver behavior analysis and detection, which are used to classify the current state of the driver. For instance, if the EEG rate of the driver is below 8 Hz, then the driver is found to be in drowsy state; a variation in the physiological features of the driver results in an emergency state being detected, the state of distraction is detected with the help of a camera installed inside the vehicle. Speeding is detected from vehicle speed with respect to traffic conditions. Bad pedaling refers to improper usage of the accelerator and brakes, which can be detected from the angle measured from the yaw angle rate. Likewise, these parameters contribute to the detection of the driver's current state. Driver behavior is analyzed in a periodical manner having a small interval of time. Figure 3 depicts the working of the A3C algorithm for the purpose of analyzing driver behavior. The states are given as input to the A3C algorithm to provide output with six types of classes in which drowsiness occurs due to continuous driving for a long period. Drivers are distracted due to abnormal activities or any events (i.e., accidents) occurs on the road. Partial conscious driving results in speeding, bad steering, and bad pedaling. Emergency situations are identified based on speed and steering behavior.

**Table 2.** Input Parameters.

| Feature Type | Feature Name | Range |
|:---:|:---:|:---:|
| Physiological | Pulse rate | 50–110 (ppm) |
| | EEG rate | 8–13 Hz |
| | Respiratory rate | 12–20 (bpm) |
| | Rhythm | Yes |
| | Q wave | 0.15 |
| | ST elevation | 0.13 |
| | ST depression | 0.16 |
| Vehicular | Speed | 40–80 km |
| | Longitudinal acceleration | 0.1–0.22 m/s$^2$ |
| | Yaw angle rate | 20$^\circ$/s–27$^\circ$/s |
| | Position | (Latitude, Longitude) |
| | Moving direction | (North, South) |
| Environmental | Traffic density | 15 vehicles/Km |
| | Accidents | 4 incidents till now |
| | Weather condition | Sunny |

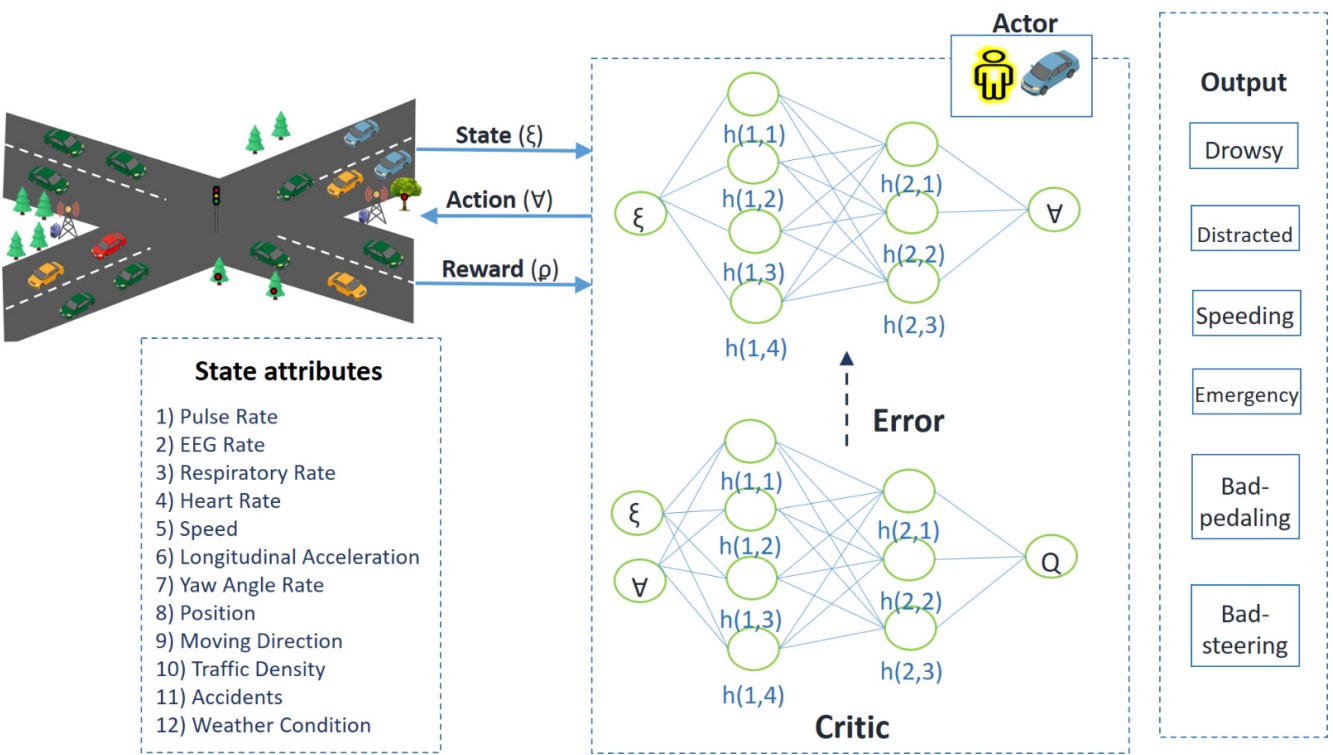

**Figure 3.** Driver Behavior Analysis Using A3C Algorithm.

*4.3. Alert Message Dissemination for Selective Regions*

The decision to disseminate alert messages is based on the current state of the driver, which is analyzed as the driver behavior [45]. The messages are disseminated to relay vehicles by implementing the JSO Algorithm on the variation in the normal range of longitudinal acceleration. Bad steering refers to inappropriate usage of steering, which is detected by sudden changes in the orientation of the basis of mobility, moving direction, and mutual distance. For all selected vehicles, a fog node sends an alert message to protect the environment. Further, the algorithm is used to detect the optimal fog node closer to the location that is in need of this information to alert the region. The jellyfish search works similar to the nature of jellyfish, which select a location for hunting food on the basis of quantity. Alert messages are disseminated on the basis of the direction of the corresponding risky vehicle, which is found as follows.

$$md = \frac{1}{n_v} \sum md_i = \frac{1}{n_v} \sum (l^* - a_t l_i) = l^* - a_t \sigma \qquad (20)$$

Here, $n_v$ is the total population around the vehicle; $l^*$ is the current best location, $a_t$ denotes the attraction factor, and $\sigma$ denotes the mean location of all vehicles. The difference between $l^*$ and $\sigma$ is $dl$, which can be formulated as follows.

$$dl = \delta \times \alpha \times r^f(0,1) \qquad (21)$$

Here, the distance of $\pm \alpha\, \delta$ is formed as the region of likelihood that messages will be disseminated, in which $\alpha$ refers to the standard deviation and can be expressed as follows.

$$\alpha = rand^\lambda(0,1) \times \sigma \qquad (22)$$

Therefore,

$$dl = \delta \times r^f(0,1) \times r^\lambda(0,1) \times \sigma \qquad (23)$$

This can be modified as follows.

$$dl = \delta \times r(0,1) \times \sigma \tag{24}$$

where

$$a_t = \delta \times r(0,1) \tag{25}$$

Hence,

$$md = l^* - \delta \times r(0,1) \times \sigma \tag{26}$$

The new position of each jellyfish is computed as follows.

$$l_i(t+1) = l_i(t) + r(0,1) \times md \tag{27}$$

Equation (30) can be modified as follows.

$$l_i(t+1) = l_i(t) + r(0,1) \times l^* - \delta \times r(0,1) \times \sigma \tag{28}$$

Here, the distributive coefficient $\delta > 0$ is related to $md$. The JSO performs two types of dissemination; one is passive, and the other is active. In passive dissemination, a message is disseminated only to vehicles in that particular cluster. In active dissemination, the message is disseminated to other vehicles on the basis of objective functions. The update location function of passive dissemination can be formulated as follows.

$$l_i(t+1) = l_i(t) + \rho \times r(0,1) \times (L_u - L_l) \tag{29}$$

Here, $L_u$ refers to the upper limit, and $L_l$ refers to the lower limit. The motion coefficient $\rho > 0$ is linked to the motion of vehicles. In active dissemination, a fog node selects another relay fog node to disseminate messages to the vehicles in order to alert the vehicles about a risk. A fog node q is selected as the relay node only if the risky vehicle moves closer to its region. The exploitation of the search space can be represented as,

$$s = l_i(t+1) - l_i(t) \tag{30}$$

where,

$$s = r(0,1) \times dir \tag{31}$$

$$dir = \begin{cases} l_q(t) - l_i(t) \ if \ f(l_i) \geq f(l_q) \\ l_i(t) - l_q(t) \ if \ f(l_i) < f(l_q) \end{cases} \tag{32}$$

Therefore,

$$l_i(t+1) = l_i(t) + s \tag{33}$$

Here, the objective function of $l$ is indicated as $f$. The switch between passive and active dissemination is controlled by a control mechanism. Initially, passive dissemination is carried out, and after a threshold time, active dissemination is started. The control mechanism comprises two values, i.e., constant $C_o$ and control function $C_t$, which takes a value between 0 and 1. Control function $C_t$ can be formulated as follows.

$$C_t = \left| \left( 1 - \frac{t}{max_{times}} \right) \times (2 \times r(0,1) - 1) \right| \tag{34}$$

Vehicles are initialized by using a logistic map and can be represented as follows.

$$l_{i+1} = \vartheta l_i (1 - l_i) \tag{35}$$

The boundary constraint for the selection of relay nodes can be formulated as follows.

$$\begin{cases} l'_{i,dim} = (l_{i,dim} - L_{u,\dim}) + L_l(\dim), \ if \, l_{i,dim} > L_{u,\dim} \\ l'_{i,dim} = (l_{i,dim} - L_{l,\dim}) + L_u(\dim), \ if \, l_{i,dim} < L_{l,\dim} \end{cases} \tag{36}$$

Here, $l_{i,dim}$ is the location of the $i^{th}$ vehicle in *dim*, which is updated as $l'_{i,dim}$ once the boundary constraint is satisfied. If there is no fog node in the region to be alerted, a virtual fog node is deployed through a constraint-based quantum entropy function to alert the vehicles in that region. The constraint for the deployment of the virtual fog node is expressed as follows.

$$F(p|q)_\varrho \overset{\text{def}}{=} F(pq)_\varrho F(q)_\varrho \tag{37}$$

where $p$ and $q$ are probabilities of entropy measures of fog deployment, and $\varrho$ represents the reward function of each virtual fog deployment. By doing so, the alert messages are disseminated to the surrounding vehicles and to the vehicles in the nearest area to optimize the traffic flow. Figure 4 illustrates the overall flow of alert message dissemination to the relay vehicles.

### 4.4. Tri-State-Aware Personalized Assistance

Each individual driver is precisely assisted on the basis of their behavior in order to avoid dangerous situations. The precise assistance is based on the past, present, and future behavior of the driver. The vehicle's current location from the GPS present in the on-board system is used for generating precise recommendations. For instance, if the driver's current state is found to be drowsy, then the location of the vehicle is obtained to assist the driver in taking a break at a nearby hotel or coffee shop. In the case that the vehicle is located in a remote area and there are no nearby locations, the recommendation will be to take a short break or to have a short nap. Assistance is provided one or more times in accordance with the current state of the driver with the multi-attribute utility model.

For example, if the current state of the driver is detected as drowsy and distracted, the model will assist, such as recommending having a cup of coffee and driving attentively. These recommendations are provided through the audio system of the vehicle; thus, the driver will be aware thereby preventing road accidents. The proposed model is significant in decision making in which the precise recommendation is provided based on the number of attributes in a certain period of time. This decision making is carried out to overcome false recommendations given under uncertain conditions. Here, recommendations are provided in both certain and uncertain situations. The utility function that represents the state is calculated for all attributes (*a*), and a decision regarding a recommendation is provided by leveraging the additive independence property which can be expressed as follows.

$$a(y_1, y_2, \ldots, y_n) = a_1(y_1) + a_2(y_2) + \ldots + a_n(y_n) \tag{38}$$

Here, $n = 1, 2 \ldots 6$ indicates the number of attributes, and the expectation (*E*) of obtaining assistance based on attributes can be represented as follows.

$$E[a(y_1, y_2, \ldots, y_n)] = E[a_1(y_1)] + E[a_2(y_2)] + \ldots + E[a_n(y_n)] \tag{39}$$

Preference toward assistance for $n$ number of attributes can be obtained as follows.

$$a(y_1, y_2, \ldots, y_n) = \sum_{i=1}^{n} h_i a_i(y_i) \tag{40}$$

Here, $h_i$ is the constant of normalization for normalizing $a$ and $a_i$ in the range of (0, 1). This assistance, along with the driver's preference and style, will get stored in the cloud for more accurate assistance. This is further used to update the A3C algorithm and AttResNet model.

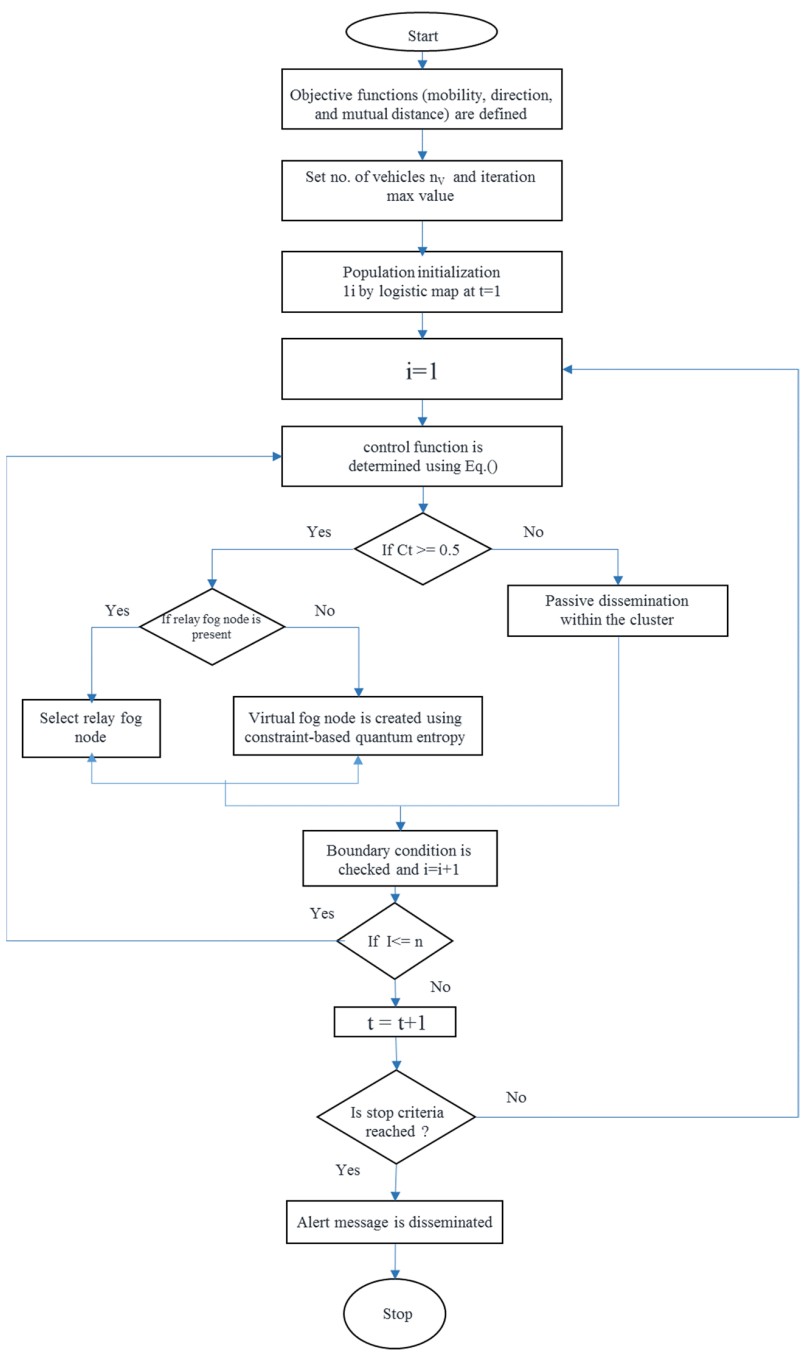

**Figure 4.** Overall Flow of Alert Message Dissemination.

## 5. Experimental Results

In this section, an experimental analysis of the proposed driver behavior analysis and personalized assistance (DBA-PA) is carried out. It comprises several sub-sections, namely simulation setup, use case, comparative analysis, and research highlights. The potentiality of the proposed research work in improving road safety and reducing road accidents is analyzed in an elaborative manner.

### 5.1. Simulation Setup

The proposed DBA-PA model was experimented on with an integration of simulation tools, that is, a simulation of urban mobility (SUMO) version 0.19.0 which is a traffic simulation based on microscopic, all the information such as number of the lane, position etc.., are simulated using SUMO [46] and of an objective modular network (OMNET++)

version 4.6, in which the prior tool is used as a traffic simulator and the latter tool is used as a network simulator. The SUMO tool is an open-source environment in which any one can implement and run their own algorithms, there are several outputs are generated for every execution of simulation. The main reason of taking SUMO tool is, it extends simulation model to consumption of fuel, and emission of noise model which supports SUMO for dense traffic scenarios. The OMNET++ also an open-source simulation time used for modeling of network which exploits discrete C++ simulator. The OMNET is reusable simulation tool and also construct network in hierarchical form that also allows simulations based on embeddings, the embeddings improve the memory management capability. The experiment was carried out by constructing a VANET network with one cloud node, four fog nodes, four E-RSU nodes, and one hundred vehicles. Table 3 presents the simulation parameters used in the experiment.

**Table 3.** Simulation Parameters.

| Parameters | Description |
|---|---|
| Network parameters | |
| Area of simulation | 2450 * 2450 m |
| Simulation time | 400 s |
| Number of cloud nodes | 1 |
| Number of fog nodes | 4 |
| Number of E-RSUs | 4 |
| Number of vehicles | 100 |
| Mobility model | Random way point |
| Range of transmission | 210–260 m |
| Type of traffic | Traffic control interface model |
| Total number of packets | 8000 (approx.) |
| Size of packet | 512 bytes |
| Transport protocol | TCP |
| Rate of transmission | 250 Mbps |
| Algorithm Parameters | |
| MCB-HCC | |
| $j$ | 20 |
| $\lambda$ | 20 |
| $\gamma$ | 0.85 |
| $\Delta$ | 0.05 |
| AttResNet | |
| $\mathbb{N}$ | 0.04 |
| Optimizer | Adam |
| $\beta_1$ | 0.91 |
| $\beta_2$ | 0.993 |
| $\varepsilon$ | $10^{-8}$ |
| Learning rate | 0.0001 |
| Drop rate | 0.2 |
| Batch size | 32 |
| A3C | |
| Discount factor | 0.97 |
| Optimizer | Stochastic gradient descent |
| Buffer size | $5 \times 10^4$ |
| Learning rate | $1 \times 10^{-4}$ |
| Batch size | 32 |
| Soft update factor | $1 \times 10^{-3}$ |
| JSO | |
| $n_v$ | 100 |
| $max_{iter}$ | 10,000 |

*5.2. Comparative Analysis*

In this sub-section, the proposed DBA-PA model is evaluated through an extensive comparison of the proposed model with other existing approaches in terms of motion prediction error, number of alerts, number of risk maneuvers, vehicle-motion-detection accuracy, driver-behavior-detection accuracy, assistance accuracy, latency, false alarm rate, safety score, and efficiency of alert message dissemination.

5.2.1. Impact of Motion Prediction Error

The motion prediction error refers to the error rate that emerges in predicting the motion of moving vehicles. Figure 5 is a comparison of the motion prediction error of the proposed DBA-PA model and DMPC [2] and TIP [35]. The motion prediction error increased as the number of moving vehicles increased. The motion prediction accuracy of the proposed model was high due to the implementation of MCB-HCC, in which the vehicles are clustered by the E-RSUs, which reduces the complexity in computation of the motion prediction process carried out by the AttResNet. The existing approaches lack in scalable computation of motion prediction.

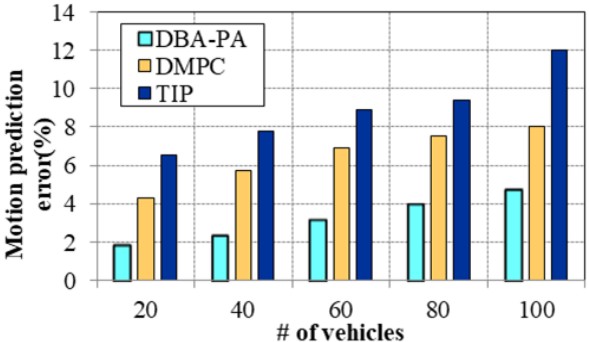

**Figure 5.** Number of Vehicles vs. Motion Prediction Error.

5.2.2. Impact of Number of Alerts

The assistance to the driver is to be provided whenever necessary, but improper analysis of driver behavior and inefficient recognition of states of danger result in an increased number of alerts to drivers, which further leads to discomfort.

Figure 6 is a comparison of the number of alerts of the proposed DBA-PA model and MADBA [29] and IDA [43] with respect to the number of vehicles. The number of alerts increased as the number of vehicles increased. The proposed model had a low number of alerts, which means that it is efficient in precisely identifying states of danger to assist the driver. This is possible due to the efficient analysis of driver behavior performed using A3C from a heterogeneous amount of data and providing alerts in accordance with it. The existing approaches lack in terms of proper detection of states of danger, which affects the performance of these approaches.

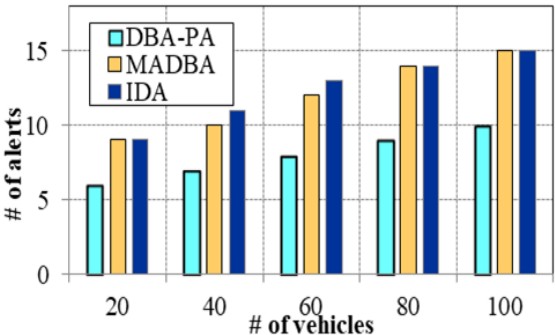

**Figure 6.** Number of Vehicles vs. Number of Alerts.

### 5.2.3. Impact of Assisted Risk Maneuvers

Risk maneuvering is defined as the process of overcoming a state of risk with the help of the assistance provided by the system. It is an important metric for measuring the efficiency of the model in terms of detection and providing assistance. Figure 7 is a comparison of the number of assisted risk maneuvers with the proposed DBA-PA model and other existing approaches with respect to the number of vehicles. The number increased as the number of vehicles increased. For instance, when the number of vehicles was 60, the number of the proposed DBA-PA model reached 17, whereas, for MADBA and IDA, the number was reduced at 14 and 12, respectively. This was due to the efficient personalized assistance provided by implementing the multi-attribute utility model in which the assistance is provided for more than one identified risk state at a time. The existing approaches lack precise generation of assistance, which limits the efficiency of these approaches.

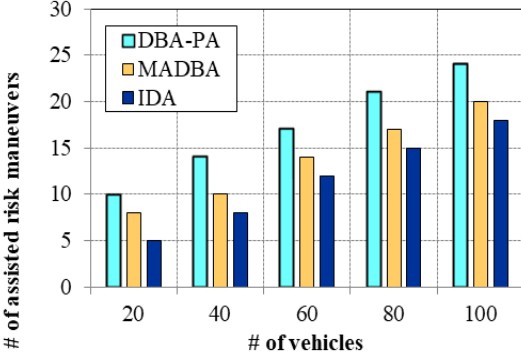

**Figure 7.** Number of Vehicles vs. Number of Assisted Risk Maneuvers.

### 5.2.4. Impact of Vehicle-motion Detection Accuracy

The detection of vehicle motion is a significant process executed to accurately analyze driver behavior in order to identify states of danger. Figure 8 depicts the accuracy of vehicle-motion detection with the proposed DBA-PA model and other existing approaches with respect to vehicle speed.

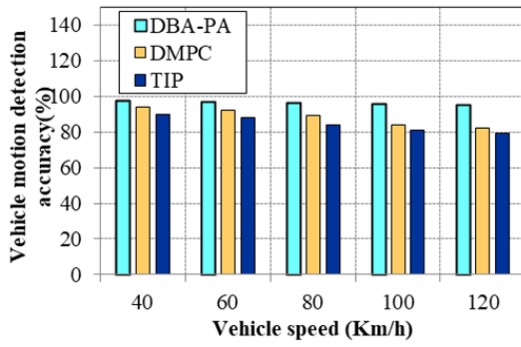

**Figure 8.** Vehicle Speed vs. Vehicle-Motion Detection Accuracy.

The vehicle-motion detection accuracy decreased as the vehicle speed increased. For example, when the vehicle was travelling at a speed of 80 Km/h, the detection accuracy of the proposed model was around 96%, whereas that of DMPC and TIP approached 89% and 84%, respectively. This is due to the formation of MCB-HCC and classification of vehicles based on mobility in each cluster with E-RSUs and AttResNet, for which mobility detection is based on velocity, speed, location, and direction. The lack of proper detection of vehicle mobility affects the performance of the existing approaches. In TIP method, Bi-LSTM was implemented to detect the behavior of vehicles based on lateral velocity, lateral acceleration and heading angle. However, Bi-LSTM takes more time for classification, and it is not able

to train huge amount of data that increases the complexity in real time environment which decreases the vehicle motion detection accuracy of this method when compared with the proposed DBA-PA method.

### 5.2.5. Impact of Driver-behavior Detection Accuracy

The accuracy of detecting driver behavior is defined as a measure for accurately detecting the behavior of the driver and classifying the driver's current state based on analysis. Figure 9 is a comparison of the accuracy of the proposed DBA-PA model and that of MADBA and HCF [31] with respect to the number of vehicles. The accuracy decreased as the number of vehicles increased; this was due to the overhead caused by the higher number of vehicles. The proposed DBA-PA model had comparatively higher accuracy than the other existing approaches due to an increase in the scalability of detection from classifying the driver's current state based on MCB-HCC and A3C. The existing approaches lack in terms of considering an increased number of vehicles in the driving environment, which results in reduced accuracy due to scalability issues.

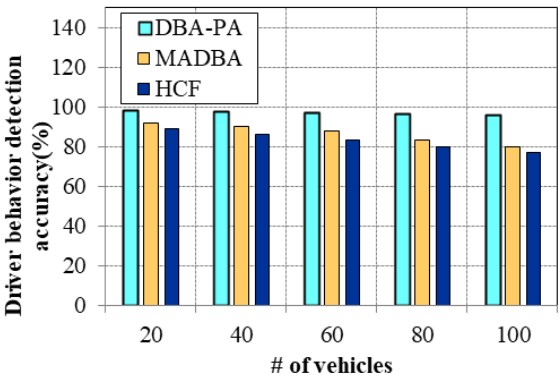

**Figure 9.** Number of Vehicles vs. Driver-Behavior Detection Accuracy.

### 5.2.6. Impact of Assistance Accuracy

The assistance accuracy is referred to as a measure for assisting the driver based on the detected behavior of the driver. For a system said to be efficient, the assistance accuracy should be high. Figure 10 is a comparison of the assistance accuracy of the proposed DBA-PA model with other existing approaches with respect to the number of risky drivers.

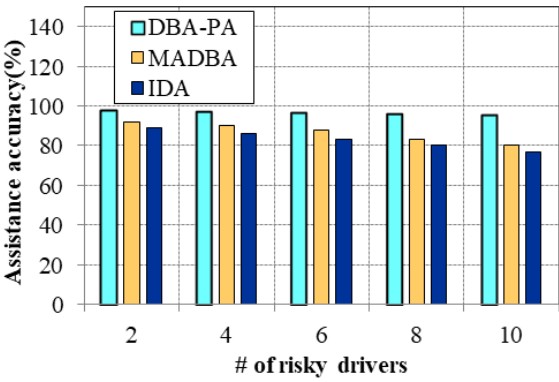

**Figure 10.** Number of Risky Drivers vs. Assistance Accuracy.

The assistance accuracy decreased as the number of risky drivers increased. The accuracy of the DBA-PA model in assisting was comparatively higher than the other approaches due to the generation of precise assistance to the drivers. This is carried out by using the multi-attribute utility model, in which the provided assistance is also based on the current location of the vehicle. Assistance is generated one or more times for drivers based

on detecting the current state multiple times to reduce the possibility of road accidents. The existing approaches provided only limited assistance, which is not appropriate for many real-time situations.

### 5.2.7. Impact of Latency

Latency is a significant factor that affects the efficiency of the behavior analysis and assistance generation system. Latency in providing assistance results in delayed assistance, which is not useful as there are many uncertainties in driving environments. Assistance must be provided with ultra-low latency for reducing casualties in the driving environment. Figure 11 is a comparison of the latency of the proposed DBA-PA model and MADBA and D-QLOA [41] with respect to the number of vehicles. It can be seen that the latency increased as the number of vehicles increased. The proposed method had low latency compared with the other existing approaches due to the computation of driver behavior and assistance generation in the fog layer. The existing approaches perform computations in a cloud server, which introduces latency in assisting, thereby affecting performance.

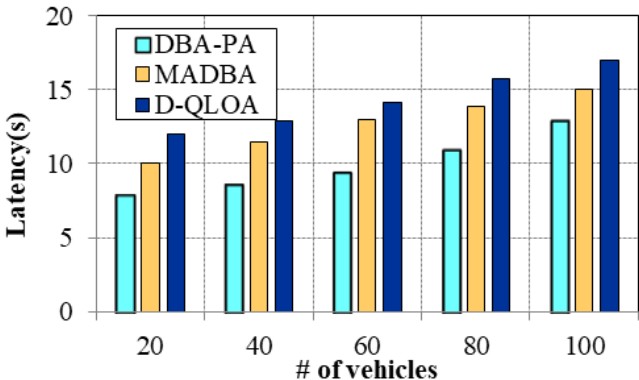

**Figure 11.** Number of Vehicles vs. Latency.

Figure 12 is a comparison of the latency of the proposed DBA-PA model and FC-IOV [7] and ACOM [28] with respect to the number of fog nodes. The latency decreased as the number increased.

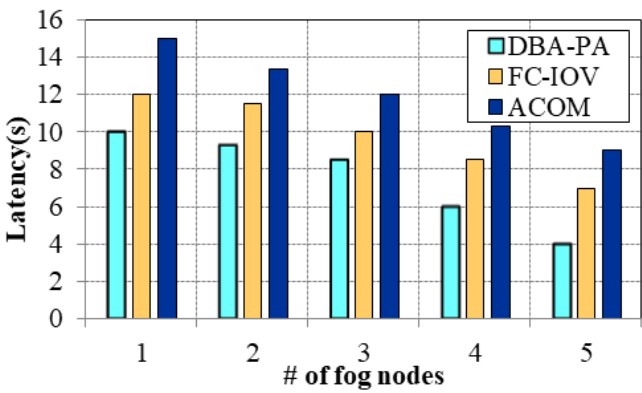

**Figure 12.** Number of Fog Nodes vs. Latency.

The proposed DBA-PA model had very low latency compared with the other approaches. This is due to the computation of behavior analysis performed using E-RSU and fog nodes. The E-RSU performs clustering and motion prediction with which the behavior analysis and prediction are carried out with ultra-low latency. The execution of clustering along with the computation of driver behavior in the fog layer reduces the latency to a very low value. For instance, the latency associated with the proposed model when five fog nodes were deployed was around 4 s, which is very low. Further, the deployment

of virtual fog nodes in the case that there are no fog nodes reduces latency to a greater extent. The existing approaches also perform computation in fog layers, but the lack of consideration for large numbers of vehicles in the environment introduces latency, which affects the proficiency of these approaches.

### 5.2.8. Impact of False Alarm Rate

The false alarm rate is due to the inaccurate detection of driver behavior and assistance provided to the driver; this further results in increased distress and panic to the driver. Figure 13 depicts the evaluation of the false alarm rate for the proposed DBA-PA model and other existing approaches with respect to number of vehicles. The false rate increased as number of vehicles increased. DBA-PA had a low rate of false alarms due to the precise detection of driver behavior and generation of personalized assistance to the driver based on behavior. For instance, the rate for DBA-PA was around 5% when there were 100 vehicles in the environment, whereas the other existing approaches had an increased rate of false alarms up to 15%. These approaches detect driver behavior by considering a limited number of parameters, which contributes to the inefficiency of these approaches.

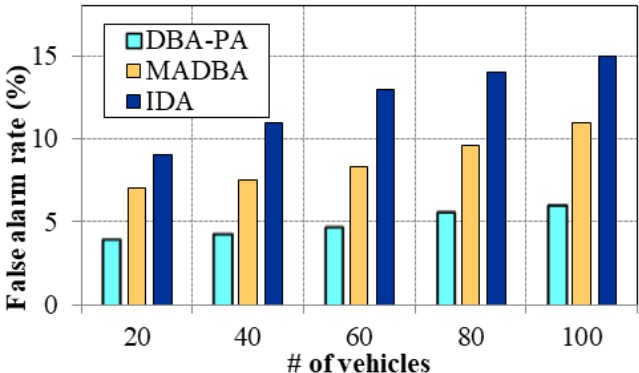

**Figure 13.** Number of Vehicles vs. False Alarm Rate.

### 5.2.9. Impact of Safety Score

The safety score is the measure of overall safety contributed by the system. It is a significant measure in calculating the quality of an approach in improving the safety of drivers and the environment.

Figure 14 is a comparison of the safety score of the proposed DBA-PA model and other existing approaches with respect to the number of vehicles. The score for DBA-PA was high compared with the other existing approaches because it contributed to improved safety for both the driver and environment. Driver safety is improved by providing accurate assistance, and the environmental safety is improved by implementing alert message dissemination, which is performed when the driver is detected as being in an emergency state. The existing approaches do not concentrate on environmental safety, which results in a reduced safety score.

### 5.2.10. Impact of Alert-Message-Dissemination Accuracy

The accuracy of alert message dissemination is a measure of how accurately alert messages are disseminated to surrounding vehicles. Figure 15 is the evaluation of the accuracy of the proposed DBA-PA model and EAC [13] and DADB [45] with respect to the number of emergency events. DBA-PA had a higher dissemination efficiency than the other approaches due to the optimal selection of surrounding vehicles both within the cluster and in the next region made in order to disseminate the alert messages. The dissemination is performed by using JSO Algorithm, which effectively disseminates the messages. If the next relay region does not have fog coverage, a virtual fog node is placed whose constraints are stated by the constraint-based quantum entropy function, thereby ensuring effective

dissemination. The existing approaches had very low efficiency in terms of dissemination due to the lack of knowledge of surrounding vehicles.

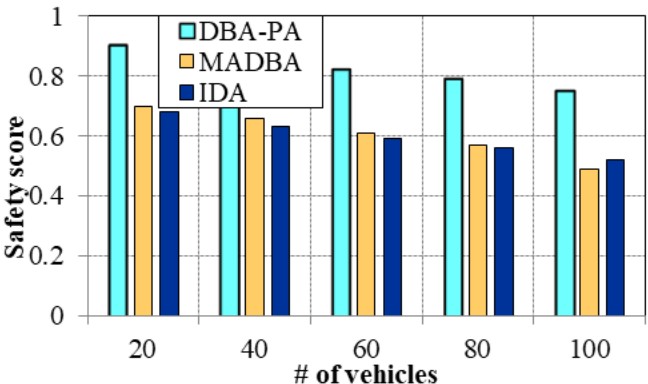

**Figure 14.** Number of Vehicles vs. Safety Scores.

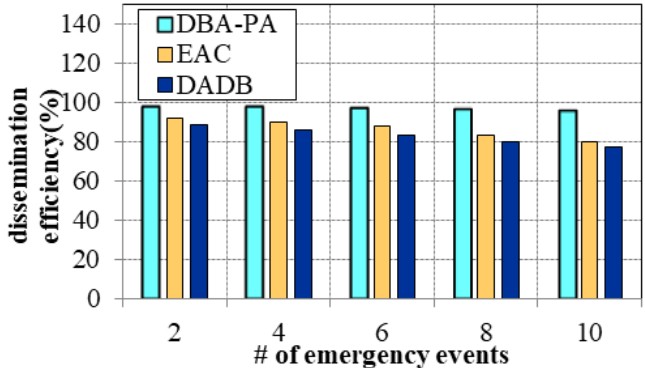

**Figure 15.** Number of Emergency Events vs. Dissemination Efficiency.

From Figures 5–15, the proposed DBA-PA model was evaluated in terms of various parameters, and it was found that it outperformed the existing approaches through significant highlights, which are stated in the next section.

*5.3. Research Highlights*

- The complexity involved in computing driver behavior dynamically is reduced by performing with MCB-HCC, which is carried out by E-RSUs. Then, the mobility of vehicles in each cluster is predicted by AttResNet, by which the vehicles are classified into three classes of mobility.
- Driver behavior is analyzed on the basis of mobility, in which the behavior of a driver is analyzed by executing A3C from several physiological features of the driver, vehicular features, and environmental features of the surrounding environment. Here, driver behavior is detected accurately and classified into several states.
- Alert messages are disseminated if the driver is found to be in a state of emergency, which is performed by the JSO Algorithm, in which the messages are disseminated in order to ensure safety to other surrounding vehicles both within the cluster and in a nearby region.
- Personalized assistance is provided to the driver on the basis of the detected current state, which is executed by the multi-attribute utility model. The assistance is provided by utilizing the current location information of the vehicle, and assistance is provided for one or more detected driver states. Instances of assistance are stored in a cloud server for training purposes.

### 5.4. Limitation and Discussion

This sub-section describes the performance discussion and limitation of the proposed DBA-PA method. Figures 5–15 illustrates the efficient performance of the proposed DBA-PA method in terms of various performance metrics. Motion prediction error (4.8%) is reduced by efficient clustering using MCB-HCC. Low number of alerts (10) is achieved by performing analysis of driver behavior using A3C Algorithm. A high number of risk maneuvers (24) is attained by efficient personalized assistance. Vehicle motion detection accuracy (95.2%) is increased by efficient clustering by MCB-HCC and vehicle classification using AttResNet by considering multiple criteria such as velocity, location, direction and speed. Increase of driver behavior detection accuracy (95.7%) by classification of driver's current state using A3C Algorithm. Improving the assistance accuracy (95.7%) by providing assistance using multi-attribute utility model. This also increases the safety score (0.75). Latency based on number of vehicles (13 s) and fog nodes (4 s) are reduced by performing generation of assistance and analysis of driver behavior in the fog layer. False alarm rate (6%) is reduced due to accurate detection of driver behavior. Alert message dissemination efficiency (95.7%) is increased by performing efficient dissemination of messages using JSO algorithm. However, this proposed DBA-PA method has addressed security issues during efficient communication between V2V and V2X. Table 4 describes the time complexity analysis for the proposed jelly fish search optimization algorithm and other algorithms which were implemented in the previous works.

**Table 4.** Analysis of Time Complexity.

| Algorithms | Time Complexity | Description |
|:---:|:---:|:---:|
| ACO Algorithm [19] | $O(n^2)$ | n represents iterations |
| Genetic Algorithm [25] | $O(gnm)$ | g denotes generation, n denotes size of population and m denotes the individuals' size |
| WO Algorithm [36] | $O(N*D)$ | N denotes population and D denotes dimension |
| JSO Algorithm | $O(n)$ | n denotes the position |

### 6. Conclusions and Future Work

In this paper, road safety in the VANET environment is improved in terms of minimizing latency, maximizing availability of resource, reducing computation overhead, and reducing delay during transmission by using the DBA-PA model. Initially, in Layer 1, E-RSUs implement MCB-HCC to form clusters of vehicles which initially computes the correlation distance between the vehicles and form as clusters to reduce the computational overhead in processing driver behavior. Then, the E-RSUs execute AttResNet to predict the motion of vehicles in each cluster by using input features such as vehicular features, environmental features, and physiological features. This information is passed to Layer 2 to analyze driver behavior. The fog node in Layer 2 analyzes driver behavior analysis by using the A3C Algorithm based on several physiological features of drivers, vehicular features, and environmental features of the surrounding environment. Driver behavior is classified into several states for which assistance is generated. If the current state of the driver is found to indicate an emergency, then an alert message is disseminated by the JSO Algorithm, where dissemination is performed for surrounding vehicles within the cluster and in neighboring regions. If a neighboring region has no fog coverage, a virtual fog node is deployed by using a constraint-based quantum entropy function. Personalized assistance is provided to the driver by using the multi-attribute utility model, in which the location characteristics of the vehicle are utilized, and assistance is provided for one or more detected driver states. From the above processes, the proposed work achieves safety ITS environment. The proposed DBA-PA model was experimented on by integrating OMNET++ and SUMO simulation tools; SUMO acts as a traffic simulator, and OMNET++

acts as a network simulator. The proposed model was evaluated by comparing it with other existing approaches in terms of motion prediction error (4.8%), number of alerts (10), number of risk maneuvers (24), vehicle-motion-detection accuracy (95.2%), driver-behavior-detection accuracy (95.7%), assistance accuracy (95.7%), latency (13 s) with respect to more number of vehicles and (4 s) with respect to more number of fog nodes, false alarm rate (6%), safety score (0.75), and alert-message-dissemination efficiency (95.7%). In the future, the proposed DBA-PA model will be improved in terms of security by implementing message encryption between V2V and V2X communication and leveraging blockchain technology for improved safety.

**Author Contributions:** Project administration, M.A.; Supervision, Y.S., M.M. (Masami MOHRI) and M.M. (Masakatu MORII); Writing—original draft, M.A.; Writing—review & editing, M.A. and Y.S. All authors have read and agreed to the published version of the manuscript.

**Funding:** This work was partially supported by the Japan Society for the Promotion of Science (JSPS) KAKENHI Grant Number JP19K11963 and JP18K04133.

**Data Availability Statement:** Not Applicable, the study does not report any data.

**Conflicts of Interest:** The authors declare no conflict of interest.

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
