# Peer review of "Three Layered Architecture for Driver Behavior Analysis and Personalized Assistance with Alert Message Dissemination in 5G Envisioned Fog-IoCV"

_futureinternet, doi:10.3390/fi14010012_

Round 1
Reviewer 1 Report
(1) Put references for AttResNet (line 109, 516 ...).
(2) Add one paragraph in the Introduction Section (line 123) to give a structure of the manuscript. Also, many existing methods and algorithms (such as MCB-HCC, AttResNet, JSO) are used but did not cite appropriately with reference numbers.
(3) Add a new 5.4 sub-section named “Limitation and Discussion” to give a limitation of the proposed method.
(4) Is it necessary to write sentences in line 82 (Heterogeneous data) and in line 87 (Uncertainties) in bold letters.
(5) Look up to line spacing in line 99. Adjust subsections with 1. and 1.2.
(6) Explain the exact classes of drivers as you mentioned in the paper’s contribution part in line 112-113.
(7) Type error in line 533.
(8) Why equations 20, 21, 22, 40, 41 42 and 43 are in bold letters and others are not?
(9) You used A3C algorithm in many places with different approaches. Like in Layer#2. It is mentioned but implementation stage in not included. Like in with part it is being used, explanation. Where its correlation with nearest fog detection.
(10) For with algorithm driver behavior classified like in output FIGURE 3? Class classification explanation. Like why driver is drowsy, for which reasons?
(11) Are you sure about the equation t=t+1 in FIGURE 4?
(12) MetaHeuristic Algorithm---jellyfish search optimization (JSO) algorithm is used to disseminate warning information to surrounding vehicles. However, the time superiority of the search algorithm and other algorithms (genetic algorithm, ant colony algorithm, simulated annealing algorithm, particle swarm algorithm, etc.) is not discussed.
(13) The author used the attention-based residual network to classify vehicles, but did not compare its performance with other neural networks in the experiment.
(14) The author proposes a novel vehicle-to-vehicle early warning model based on driving behavior analysis using 5G network, which combines search algorithms and deep learning algorithms. It is a good idea. However, in the discussion, the issue of time delay was not discussed. Combining 5G for cloud to base station and then to the data distribution of vehicles, in reality, the 5G network signal and delay is still a problem, I hope the author can consider it.
(15) The author uses pulse rate, EEG rate, respiration rate, and heart rate as a reference for driving behavior in the analysis of driver behavior. Whether there are problems such as repeated information collection and non-essential data collection, and how to confirm the accuracy of the data?
Author Response
Dear Reviewer 1
Thank you for your time and comments
Please see the attachment for our response

Reviewer 2 Report
In the present manuscript, the authors propose a driver behavior analysis and personalized assistance to drivers and to alert drivers of casualties around them as a step to preventing them from harm. Therefore, I will mention some aspects to improve the quality of the article:
-The authors use the acronyms incorrectly, since the correct form is “Intelligent Transportation Systems (ITS)”. Those errors must be reviewed and fixed in all acronyms of the manuscript. Also, all acronyms must have their corresponding meaning.
-Authors must use the MDPI manuscript format.
-It is not necessary to make subsections in the Introduction Section.
-Be careful with the titles of the Sections, because all their letters must not be written in capital letters.
-Avoid using phrasal verbs in the manuscript.
-In line 144, there is a double comma.
-You can add these articles as a reference to improve the quality of the manuscript:
--Zambrano-Martinez, J. L., Calafate, C. T., Soler, D., Cano, J. C., & Manzoni, P. (2018). Modeling and characterization of traffic flows in urban environments. Sensors, 18(7), 2020.
--Behrisch, M.; Bieker, L.; Erdmann, J.; Krajzewicz, D. SUMO—Simulation of urban mobility: An overview. In Proceedings of the Third International Conference on Advances in System Simulation. ThinkMind (SIMUL 2011), Barcelona, Spain, 23–28 October 2011; IARIA XPS Press: København, Denmark, 2011.
- There are objects such as Tables, Figures, Equations that are not being mentioned in the manuscript, or are too far where they have been cited. In addition, they must be written completely and with their first letter in capital letters.
-The algorithm that is in line 467, needs to be written in a better form and that the information is not being mounted.
-Instead of using the word pseudo code, use better algorithm.
-There are terms of the Equations that are not with their respective explanation.
-Figure 1, the letters are too small. It is not possible to have a good visualization.
- Are the data obtained for the simulation real or synthetic?
-The proposal presented by the authors, when simulated in a real environment, what would be the possible changes that would exist?
-What is the reason that the simulation was performed with 100 vehicles and in a time of 900s?
-What happens if the number of vehicles is gradually increased and there is traffic congestion?
-In the simulation, is it obviating the different obstacles that may exist in the real world such as buildings, trees, etc. for a correct communication to the vehicle?
-What versions of the simulators are authors using?
-The authors have not posted a brief introduction to the simulators they used.
-The numbering of the manuscript sheets is incorrect.
-The conclusions obtained must be improved.
Author Response
Dear Reviewer 2
Thank you for your time and comments
Please see the attachment for our response

Round 2
Reviewer 1 Report
- Recommend to review carefully before final version. Thanks.
Author Response
Thank you for reviewing our manuscript and grateful to your kind response, We really appreciate your constructive comments and instructions.
We tried to improve the manuscript more
Reviewer 2 Report
Thanks to the authors for performing some of the changes suggested by the reviewers. I have observed that authors have used the corresponding format of the magazine. However, to present to the reviewers, this manuscript must be done with the latest changes performed by the authors, and not with those boxes that appear on the right side of the manuscript, which is a terrible error on the part of the authors when presenting a scientific article with those terrible flaws. It is advisable to detail in the Cover Letter what changes have been performed, and in the manuscript highlight where the changes have been performed. Therefore, I detail the changes that have not been performed or new mistakes that the authors have:
-There is no ascending number in the bibliographic citations in the text.
-There are bibliographic references that have not been cited in the manuscript.
-The authors must place the answers to the questions asked in the article.
- All acronyms are not spelled correctly.
-When writing "Section", "Algorithm", "Figure", "Table", "Equation", in a scientific article, it must always be with the first capital letter and with the complete word.
-The titles of the Sections and Subsections are not in order.
-Algorithm 1 has the line number superimposed.
-There is an "i" on line 760.
-The text on line 717 has been cut.
-The Conclusions Section must be titled "Conclusions" and must be numbered in ascending order.
Author Response
Thank you for reviewing our manuscript and grateful to your kind response, We really appreciate your constructive comments and instructions. and very sorry for the mistake we made last time.
We have revised this manuscript according to your requirements and suggestion. We marked the new manuscript in red color.